# Factors influencing the practice of Smoking Cessation Assessment and Management among Primary Care Doctors (SCAAM-DOC) in three districts of Malaysia

**Beatrice Jee Ngee Ling** [1]*, **Ai Theng Cheong** [2]*, **Abdul Hadi Abdul Manap** [2]

**1** Bandar Baru Health Clinic, Ministry of Health Malaysia, Kuala Langat, Selangor, Malaysia, **2** Department of Family Medicine, Faculty of Medicine and Health Sciences, Universiti Putra Malaysia, Serdang, Selangor, Malaysia

* beatricejee.moh@gmail.com (BJNL); cheaitheng@upm.edu.my (ATC)

**Data Availability Statement:** The data contains potentially identifying healthcare providers' information such as; age, gender, years of service, occupation and smoking status in the study

## Abstract

### Background

Smoking prevalence remains high in Malaysia. Primary care doctors have a good opportunity to motivate the smokers to quit smoking in view of the accessibility of primary healthcare clinics to the public. The objective of this study was to determine the practice of smoking cessation management among primary care doctors and its associated factors.

### Methods

A cross-sectional online survey was carried out among 383 medical officers and interns in all government primary healthcare clinics in the district of Petaling, Klang and Hulu Langat from June to August 2020. All doctors were involved in the care of patients for smoking cessation. The knowledge, attitude and practice of smoking cessation management were assessed using a 17-items validated questionnaire which covered the components of 5As (Ask, advise, assess, assist, arrange) and 5Rs (Relevance, risk, reward, roadblocks, repetition). The management of pre-contemplation phase included the components of ask, advise, assess and 5Rs. The management of the contemplation phase included the components of assist and arrange.

### Result

The majority of the respondents had poor score of knowledge (62.4%); attitude (58%) and practice (pre-contemplation management:50.9%; contemplation management:75.7%). Using multivariate logistic regression analysis, the significant factors associated with the poor practice of smoking cessation management in the pre-contemplation phase were poor (OR = 2.14, 95% CI 1.11–4.12, p <0.01) or moderate knowledge (OR = 2.50, 95% CI 1.19–5.26, p<0.01), poor attitude (OR = 2.16, 95% CI 1.39–3.37, p<0.01), lacks smoking cessation banners, brochures and leaflets in the clinic (OR = 2.01, 95%CI 1.26–3.19, p<0.01) and

population in the three districts. Contact information for a data access: Medical Research and Ethics Committee, National Institutes of Health, Ministry of Health Malaysia, Block A, Level 2, No 1, Jalan Setia Murni U13/52, Seksyen U13, Setia Alam, 40170, Shah Alam, Selangor, Malaysia Telephone: +603 – 3362 8205 mrecsec@moh.gov. my.

**Funding:** The authors received no specific funding for this work.

**Competing interests:** The authors have declared that no competing interests exist.

lack of nicotine replacement medications (OR = 2.27. 95%CI 1.27–4.06, p<0.01). No significant factors were shown associated with the practice of the contemplation phase.

## Conclusion

The majority of primary care doctors had poor knowledge, attitude and practice of smoking cessation management. Factors that had increased the odds of the poor practice of smoking management at the pre-contemplation phase were poor knowledge, poor attitude, and insufficient organizational support for health promotion materials and nicotine replacement medication.

## Introduction

Smoking carries high mortality and causes approximately 90% of all lung cancer deaths, 80% of all chronic obstructive pulmonary disease deaths and increases the risk for death from all causes in men and women worldwide [1]. In Malaysia, it was estimated that 20,000 deaths annually were related to tobacco [1, 2]. In Malaysia, the National Health and Morbidity Survey 2019 reported that the prevalence of smokers aged 15 years and above was 21.3% and this contributed to about 4.8 million smokers [3]. The proportion of smokers was 30 times higher among males compared to the females [40.5% (95% CI: 37.90, 43.06) vs 1.2% (95% CI: 0.85, 1.70)] [3].

For curbing tobacco use in Malaysia, apart from supporting the World Health Organization Framework Convention on Tobacco Control and implementing legislation on local tobacco production via Control for Tobacco Products Regulation, Malaysia had taken the initiative to set up smoking cessation services (mQuit services) in most government primary healthcare clinics and hospitals [4, 5]. The mQuit services incorporate behavioural and pharmacological approaches in smoking cessation management according to the national clinical practice guidelines on tobacco disorder [6].

The health care system in Malaysia is provided by both the public and private health sectors, and the Ministry of Health being the major provider for the public sector [7, 8]. The government healthcare clinics are highly subsidized and accessible to the public [7, 8]. The patient only needs to pay RM1 to RM5 (USD 0.30–1.20) for a clinic visit [7, 8]. This fee includes consultation, investigations and medications [7, 8]. The doctors in government healthcare clinics encounter more chronic disease follow-up cases while in private primary care clinics more acute illnesses were seen [9]. The mQuit services incorporated in the government primary healthcare clinic services are a good effort in view of their accessibility and affordable charges [7, 8]. This service would benefit and facilitate those smokers who intend to quit smoking [7, 8].

Studies had shown that advice from health professionals was effective in increasing cessation, primarily through aiding and motivating them to make a quit attempt [10]. It was found that smokers were 1.66 times more likely to quit smoking with brief advice than no advice [10]. However, literature has showed that the practice of smoking cessation management among primary care doctors were very sparse [11–14]. This could be due to suboptimal knowledge and attitude among them [12, 14]. Previous studies showed that doctors with good knowledge and attitude were significantly associated with good practice of smoking cessation management [12, 14–16]. Thus, this study aimed to determine the level of practice of smoking cessation management among primary care doctors and its associated factors. We

hypothesized that the prevalence of practice of smoking cessation management among primary care doctors would be low and there was an association between socio-demographic factors, organizational support, level of knowledge, attitude with the practice of assessment and management in smoking cessation in the pre-contemplation and contemplation phase among primary health care doctors. It is hoped that the results of this study would identify the areas that need to be targeted for further improvisation of the practice of smoking management among the primary care doctors.

## Materials and methods

### Study design and data collection

A cross-sectional study was conducted in all government primary healthcare clinics in the district of Hulu Langat, Klang and Petaling in the state of Selangor. These three districts were purposely selected in view of the high density of the population and the large number of doctors served in these areas. There was a total of 32 government primary healthcare clinics comprised of 223 doctors in the district of Hulu Langat, 132 doctors in the district of Klang and 198 in the district of Petaling. The study was conducted from June to August 2020. All the doctors were involved in identifying and counselling smokers to quit smoking during their daily consultation and recruiting those in the contemplation phase to quit smoking clinic.

All primary care doctors consisted of the medical officers and interns working at the government primary healthcare clinics in the district of Hulu Langat, Klang and Petaling were invited to participate in the study. Medical officers are doctors who had passed two years of internship training and are involved in the clinical management of patients in primary healthcare clinics while interns are medical graduates who are still undergoing internship training under the supervision of a specialist. Those on long leave for more than one month were excluded.

The questionnaire was an online questionnaire (refer S1 File). Anonymous of the participants of the survey had been carried out to ensure confidentiality and mitigate response bias. This could encourage participants to answer the questionnaire honestly reflecting their practice. There were three sections in this questionnaire, whereby the first section examined the socio-demographic characteristics of the primary care doctors (age, gender, years of experience, position of occupation and smoking status). This section also examined the number of smokers encountered by the respondent in the past month during a routine clinic consultation and the number of smokers who were willing to quit smoking. The second section examined the organizational support, including the availability of health promotion materials (smoking cessation banners, brochures and leaflets), designated quit smoking clinic services, training or courses for smoking cessation, and nicotine replacement medications. The third section examined the knowledge, attitude and practice of primary care doctors, using a locally validated questionnaire.

This validated questionnaire consisted of four component (knowledge, attitude, practice of management in the pre-contemplation phase and practice of management in the contemplation phase) [17]. There were total of 17 items with 2 items on knowledge, 10 items on the practice of smoking cessation assessment and management at the pre-contemplation phase, 2 items on the practice of smoking cessation assessment and management in the contemplation phase and 3 items on attitude [17]. These 17 items have demonstrated a good Cronbach's alpha value for all four component. The value of Cronbach's alpha was 0.84 (the practice of smoking cessation assessment and management at the pre-contemplation phase), 0.74 (the practice of smoking cessation assessment and management in the contemplation phase), 0.80 (knowledge) and 0.60 (attitude) [17]. This questionnaire was validated among 141 primary

care doctors from the government health clinics in three districts in the state of Pahang in Malaysia [17].

The assessment of the knowledge component was on the 5A components ('Ask', 'Assess', 'Advice', 'Assist', 'Arrange') [17]. The two questions mainly assessed the primary care doctors' familiarity with the sequence and components of the 5As. (Refer S1 File) These were assessed with a dichotomous scale consisting of options of 'True' and 'False' [17]. A score of one was given for a correct answer and 0 score for a wrong answer [17].

The attitude component was assessed according to the doctors' perception on the relevancy of clinical practice guidelines in improving smoking cessation, the worthiness of putting effort in helping smokers to quit smoking and the benefit of repetition in giving advice to assist patients in smoking cessation [17]. It was assessed according to the respondents' degree of agreement with a five point Likert scale of 'Strongly agree', 'Agree', 'Don't know', 'Disagree' and 'Strongly disagree' [17]. The scores were given as 'Disagree' = 1 mark, 'Strongly disagree' = 2 marks, and other responses (strongly agree, agree, don't know) = 0 mark and for the final item about 'repetition in advising to quit smoking is beneficial'; the reverse coding was applied with the score of 'Agree' = 1 mark, 'Strongly agree' = 2 marks and other responses (strongly disagree, disagree, don't know) = 0 mark (Refer S1 File) [17].

The practice component was assessed according to the stage of change: the precontemplation and contemplation phase [17]. The practice of smoking cessation assessment and management for patients at pre-contemplation phase include 'Ask', 'Advice' and 'Assess', followed by the 5R on those who are not ready to quit [17]. The practice of smoking cessation management for those patients at contemplation phase include 'Assist' and 'Arrange'. The items were assessed using a four point Likert scale with the score of 'Always' = 2 marks, 'Frequent' = 1 mark, other responses (seldom and never) = 0 mark [17].

## Ethical issues

Ethical approval for this study was obtained from the Medical Research and Ethics Committee (MREC), Ministry of Health Malaysia (NMRR-19-2175-48947).

## Data analysis

The data was undertaken using the IBM SPSS statistic version 26.0. There were two outcomes in this study: the components of the practice of smoking cessation management in the pre-contemplation phase and the practice of smoking cessation management in contemplation phase. The median score of these outcome variables was reported as the data was not normally distributed. The practices were reclassified into two groups, the good practice group for those who scored higher than the median score and the poor practice group for those who scored equal to or less than the median score.

The total score for knowledge was 2. It was categorized as poor knowledge (score-0), average knowledge (score-1) and good knowledge (score-2). The attitude score ranged from minimum score of 0 to maximum score of 6. Those scored higher than median score was classified as good attitude and those scored equal or less than the median score was classified as poor attitude.

To examine the associated factors with the practice of smoking cessation assessment and management at the pre-contemplation and contemplation phase, Pearson Chi-Square/Fisher exact test was used for bivariate analysis and multiple logistic regression was used for multivariate analysis. From the bivariate analysis, factors with p-values equal to or less than 0.25 were included in the multiple logistic regression.

Testing for multicollinearity, assumptions and outliers was also carried out before multiple logistic regression analysis. Testing for multicollinearity of the independent variables was carried out by examining the variance inflation factor (VIF). There was no multicollinearity detected and the VIF ranged from 1.01 to 6.88. The tolerance level of 0.1 (= VIF 10) was used. The statistical significance in the final model was accepted at p-values equal to or less than 0.05. The model fitness was assessed using the Hosmer-Lemeshow goodness of fit test. The analysis with Hosmer-Lemeshow test showed a p-value of more than 0.05, indicating an adequate model fit. The outliers were checked using the Cook's distance, leverage value, studentized and standardized residual and the values were within the acceptable limit.

## Results

The response rate was 69.2% (383/553). Most of the participants (56%) were aged 31 to 35 years, and the median age was 33 (IQR 4) years. The majority of the respondents (71.5%) was female. More than half (56.1%) of the respondent had 6 to 10 years of service experience, and the median years of service was 8 (IQR: 5) years. The majority of the respondents were medical officers (98.4%), and reported that they had never smoked (96.9%) (Refer Table 1).

The majority of the respondents (92.2%) had encountered smokers during consultation in the past one month, and 76.2% of these respondents who had encountered smokers reported that those smokers were willing to quit smoking. The majority 84.9% (325/383) of the primary health care doctors reported a designated smoking cessation clinic at their clinic, and 87.2% (334/383) reported that they had attended smoking cessation courses.

Meanwhile, 64.2% (246/383) of primary health care doctors reported the unavailability of smoking cessation banners, brochures and leaflets and 78.1% (299/383) reported the unavailability of nicotine replacement medications.

The median score for the attitude was 2 (IQR 2) and practice for the pre-contemplation and contemplation phase was 8 (IQR 5) and 2 (IQR 1) respectively. More than half of the doctors

**Table 1. Sociodemographic characteristics of respondents (N = 383).**

| Characteristic | Frequency | Percentage |
|---|---:|---:|
| **Age** | | |
| ≤30 years | 85 | 22.2 |
| 31 to 35 years | 215 | 56.1 |
| 36 to 40 years | 57 | 14.9 |
| ≥41 | 26 | 6.8 |
| **Gender** | | |
| Male | 109 | 28.5 |
| Female | 274 | 71.5 |
| **Years of service** | | |
| ≤5 years | 99 | 25.9 |
| 6 to 10 years | 215 | 56.1 |
| ≥11 years | 69 | 18.0 |
| **Occupation** | | |
| Medical officer | 377 | 98.4 |
| Intern | 6 | 1.6 |
| **Self-reported smoking status** | | |
| Never smoker | 371 | 96.9 |
| Former smoker | 10 | 2.6 |
| Current smoker | 2 | 0.5 |

attained poor scores in the components s of knowledge, attitude and practice. There was a preponderance of 62.4% (239/383) of doctors with a poor knowledge score of 0. Most doctors (58%) had a poor attitude (total score ≤2). Half of the doctors (50.9%) reported having poor practice (total score ≤8) in the pre-contemplation phase and 75.7% of doctors had a poor practice of smoking cessation assessment and management (total score ≤2) in the contemplation phase (Refer Table 2).

For the knowledge component, more than half of doctors (62.4%) attained the wrong answer for both questions. 77.5% and 68.4% of the participants attained wrong answers for the 'assess' and 'assign' components of smoking cessation management respectively. In the attitude component, a small proportion (6.5% to 14.4%) of doctors obtained the full score of 2 for all three questions whereby only 6.5% disagreed on their effort in helping smokers to quit was not well rewarded while 14.4% of them disagreed on clinical practice guidelines was not relevant in smoking cessation management. Meanwhile, only 15.4% of the doctors agreed that repetition in advising smokers was beneficial.

For the practice of smoking cessation management in the pre-contemplation phase, only about one-fifth (17%) of the doctors always assessed the patients' status of smoking and about 40% of them always advised patients on quitting smoking. For the practice of smoking cessation management at the contemplation phase, there was only one-tenth of the doctors (13.1%) had always provided smokers with practical counselling. For the 5R components, the result generally showed a poor practice as less than a quarter of doctors had always practised 'Relevance' (19.1%); 'Risk' (14.1%); 'Reward' (15.1%); 'Road-blocks' (14.1%) and 'Repetition' (26.1%) (Refer Table 3).

Four factors: (1) Organizational support of quit smoking promotion materials (availability of smoking cessation banners, brochures and leaflets (2) Availability of nicotine replacement medication (3) Knowledge) and (4) Attitude were significantly associated with the practice of smoking cessation management at the pre-contemplation phase (refer Table 4). Primary health care doctors who reported a lack of organizational support for quit smoking promotion materials had 2.01 times higher odds of poor practice (OR = 2.01, 95%CI 1.27, 3.20, p<0.01) compared to those that have these in their clinic. Primary health care doctors who reported a lack of nicotine replacement medications in their clinic had 2.28 times higher odds of having poor

**Table 2. The level of knowledge, attitude and practice of smoking cessation management among primary care doctors.**

| Variable | Frequency | Percentage |
|---|---|---|
| Knowledge (total score) | | |
| Poor (0) | 239 | 62.4 |
| Average (1) | 81 | 21.1 |
| Good (2) | 63 | 16.5 |
| Attitude (total score) | | |
| Poor (0–2) | 222 | 58.0 |
| Good (3–6) | 161 | 42.0 |
| Practice at the pre-contemplation phase (total score) | | |
| Poor (0–8) | 195 | 50.9 |
| Good (9–20) | 188 | 49.1 |
| Practice at contemplation phase (total score) | | |
| Poor (0–2) | 290 | 75.7 |
| Good (3–4) | 93 | 24.3 |

IQR = Interquartile Range.

**Table 3. The likeliness of practice among primary care doctors for the items of smoking cessation management in the pre-contemplation and contemplation phase.**

| Items | Pre-contemplation phase | | | | | | | |
|---|---|---|---|---|---|---|---|---|
| | Never | | Seldom | | Frequent | | Always | |
| | n | % | n | % | n | % | n | % |
| I will check when the last time is that my patient smoked (Ask) | 14 | 3.6 | 118 | 30.8 | 186 | 48.6 | 65 | 17.0 |
| I advise the smokers to quit (Advise) | 0 | 0.0 | 31 | 8.1 | 212 | 55.3 | 140 | 36.6 |
| I advise the smokers to reduce amount of cigarettes per day (Advise) | 2 | 0.5 | 50 | 13.1 | 202 | 52.7 | 129 | 33.7 |
| I inquire the smoker's willingness to quit (Assess) | 2 | 0.5 | 160 | 41.8 | 160 | 41.8 | 61 | 15.9 |
| I encourage the smokers to indicate why quitting is personally important (Relevance) | 2 | 0.5 | 121 | 31.6 | 187 | 48.8 | 73 | 19.1 |
| I ask the smokers to identify any potential harm to self from smoking (Risk) | 18 | 4.7 | 135 | 35.2 | 176 | 46.0 | 54 | 14.1 |
| I ask the smokers to identify negative consequences of continuing smoking (Risk) | 14 | 3.7 | 130 | 33.9 | 183 | 47.8 | 56 | 14.6 |
| I ask the smokers to identify the advantages of quit smoking to their family (Reward) | 13 | 3.4 | 131 | 34.2 | 181 | 47.3 | 58 | 15.1 |
| Ask smokers to why quitting is impossible (Roadblock) | 11 | 2.9 | 152 | 39.7 | 166 | 43.3 | 54 | 14.1 |
| I continuously inform the smokers benefit of quit smoking (Repetition) | 0 | 0.0 | 81 | 21.2 | 202 | 52.7 | 100 | 26.1 |
| | Contemplation phase | | | | | | | |
| I provide the smokers with practical counselling (Assist) | 11 | 2.9 | 153 | 39.9 | 169 | 44.1 | 50 | 13.1 |
| I give further follow-ups for smokers quitting (Arrange) | 30 | 7.8 | 41 | 10.7 | 201 | 52.5 | 111 | 29.0 |

practice (OR = 2.28, 95%CI 1.28, 4.06, p<0.01) compared to those who had nicotine replacement medications in their clinic. Those with poor and moderate knowledge had 2.14 times odds (OR = 2.14, 95%CI 1.12, 4.12, p = 0.02) and 2.51 times odds (OR = 2.51, 95%CI 1.19, 5.27, p = 0.02) respectively to have poor practice compared to those who had good knowledge. Primary health care doctors with poor attitude had 2.17 times odds to have poor practice (OR = 2.17, 95%CI 1.39, 3.38, p<0.01) compared to those who had good attitude.

For the binary logistic regression analysis of the practice at the contemplation phase of smoking cessation management, there were no significant factors identified (refer S2 File).

## Discussion

We found that the proportion of primary care doctors with poor knowledge, attitude, and practice of smoking cessation assessment and management was high: 62.4% for knowledge, 58% for attitude, 50.9% for practice at pre-contemplation and 75.7% for practice at contemplation phase. Many of them reported the unavailability of smoking cessation banners, brochures, leaflets (64.0%) and nicotine replacement medications (78.0%) in their clinics. Factors significantly associated with the practice of smoking cessation management at the pre-contemplation phase were organizational support of health promotion materials (availability of smoking cessation banners, brochures and leaflets), availability of nicotine replacement medication, knowledge and attitude of primary care doctors.

The burden of the smoking problem in Malaysia is high [3]. In our study, 92.2% of the primary care doctors had encountered smokers during consultation in the past one month, and 76.2% of these doctors who had encountered smokers reported that those smokers were willing to quit smoking. Good knowledge, attitude and practice among doctors are essential to aid these smokers in quitting smoking effectively [17]. However, this study generally showed poor scores in all three knowledge, attitude and practice components among primary care doctors. This result was similar to the local study conducted among primary health care doctors in the state of Pahang [12].

Despite most primary care doctors reported attending training (87.2%) for smoking cessation in our study, more than half (62.4%) of the doctors were found to have poor knowledge. This was consistent across other studies among doctors despite disparate tools and knowledge

**Table 4. Factors associated with the practice of smoking cessation management at precontemplation phase among primary health care doctors.**

| | Preliminary model (SLR) | | | | Final model (MLR) | | | |
|---|---|---|---|---|---|---|---|---|
| | COR | 95% CI | | p- value | AOR | 95% CI | | p- value |
| | | Lower | Upper | | | Lower | Upper | |
| **Gender** | | | | | | | | |
| Female | 1.00 | | | | 1.00 | | | |
| Male | 0.58 | 0.37 | 0.91 | **0.02** | 0.60 | 0.36 | 1.00 | 0.05 |
| **Years of service** | | | | | | | | |
| ≥11 years | 1.00 | | | | 1.00 | | | |
| 6 to 10 years | 1.42 | 0.82 | 2.47 | **0.21** | 0.92 | 0.49 | 1.70 | 0.78 |
| ≤5 years | 2.35 | 1.25 | 4.41 | **0.01** | 1.19 | 0.58 | 2.44 | 0.64 |
| **Occupation** | | | | | | | | |
| Medical officer | 1.00 | | | | 1.00 | | | |
| Intern | 4.92 | 0.57 | 42.52 | **0.15** | 2.09 | 0.21 | 21.22 | 0.53 |
| **Smoking status** | | | | | | | | |
| Never smoker | 1.00 | | | | 1.00 | | | |
| Former smoker | 4.02 | 0.84 | 19.19 | **0.08** | 1.34 | 0.26 | 7.03 | 0.73 |
| **Organization support:** | | | | | | | | |
| **Smoking cessation banners, brochures and leaflets** | | | | | | | | |
| Yes | 1.00 | | | | 1.00 | | | |
| No | 1.33 | 1.55 | 3.66 | **<0.01** | 2.01 | 1.27 | 3.20 | **<0.01** |
| **Nicotine replacement medication** | | | | | | | | |
| Yes | 1.00 | | | | 1.00 | | | |
| No | 2.90 | 1.73 | 4.86 | **<0.01** | 2.28 | 1.28 | 4.06 | **<0.01** |
| **Knowledge** | | | | | | | | |
| Good = 2 | 1.00 | | | | 1.00 | | | |
| Average = 1 | 2.62 | 1.31 | 5.24 | **<0.01** | 2.51 | 1.19 | 5.27 | **0.02** |
| Poor = 0 | 2.91 | 1.60 | 5.27 | **<0.01** | 2.14 | 1.12 | 4.12 | **0.02** |
| **Attitude** | | | | | | | | |
| good>2 | 1.00 | | | | 1.00 | | | |
| Poor ≤2 | 2.09 | 1.38 | 3.15 | **<0.01** | 2.17 | 1.39 | 3.38 | **<0.01** |

SLR: Simple logistic regression.

MLR: Multiple logistic regression.

95% CI: 95% confidence interval.

COR: Crude odd ratio.

AOR: Adjusted odd ratio.

components being assessed [11, 14, 18, 19]. Also, consistent with other studies, our result demonstrated that primary care doctors with poor knowledge were significantly associated with poor practice [10, 14–16]. This discrepancy needs to be explored further to explain the poor knowledge scores despite the majority of our primary care doctors having attended training courses for smoking cessation management. Perhaps the refresher course is needed and the training courses might need to be improvised to increase their knowledge and confidence in the delivery of smoking cessation management efficaciously.

The majority of primary care doctors (73.8% to 93.3%) in Saudi Arabia had shown a good attitude in smoking cessation assessment and management [11, 13]. Similarly, healthcare providers in China, Egypt, Africa, and Saudi Arabia had shown a good attitude in smoking cessation assessment and management [10, 16, 20–23]. Our result showed near 60% of the primary

care doctors reported having a poor attitude. The disparity observed in the poorer attitude toward smoking cessation management in Malaysia compared to other countries showed the need for a more rounded approach to exploring this poor attitude among our primary care doctors.

A local study looking at the primary care doctors' attitude toward smoking cessation in the state of Pahang also reported a similar finding as ours [12]. However, another local study in a teaching hospital found that the majority of the interns (89–95%) showed a good attitude toward smoking cessation management [19]. The different working cultures and environments might contribute to the difference in the results seen in these settings. Similar to other studies, our study showed that primary care doctors with poor attitudes were associated with inadequate practice in smoking cessation management [10, 15]. Therefore, the poor attitude among primary care doctors would need to be addressed to increase their level of provision in smoking cessation management.

Our study found that the practice of smoking cessation management at the pre-contemplation phase and contemplation phase was poor. Literature had also shown that there was a suboptimal self-reported practice of smoking cessation management using the 5A model [10, 13, 14]. Previous studies did not probe into the practice of smoking cessation management separately in the pre-contemplation and contemplation phase, and there was only one outcome used in the literature that included all 5As as a single outcome [11, 13–15, 18, 19].

For the management at the precontemplation phase, our result was consistent with the literature which showed that the best practice was the "Ask" and "Advise" components as compared to the "Assess" component [11, 18, 19]. This result implied that, apart from occasionally screening for smoking and offering advice to quit smoking, most doctors did not go further to assess smokers' willingness to quit. This could be explained by the poor knowledge of the doctors. In addition, we found that the practice of smoking cessation management for the 5R components was also poor, as only less than a quarter of doctors had always practised these. We could not find literature which probed into these components. Nevertheless, these components are essential to be included in the assessment as it reflects doctors' competency in delivering smoking cessation management and could be a potential area to be targeted for improvement of doctors' competency in the practice of smoking cessation management.

Our study showed that the practice of smoking cessation management in contemplation phase was poorer than in the pre-contemplation phase. This could be due to the doctors might not encounter many patients who were in the contemplation phase. We could not find any significant factors that contributed to the poor practice during the contemplation phase. Further study is needed to explore the possible barriers to this.

Inconsistent with the literature, our study showed that organizational support such as the availability of health promotion materials and nicotine replacement medication were found to be significant factors associated with the practice of quitting smoking management [10, 16]. More than half of the doctors in our study reported the unavailability of smoking cessation banners, brochures and leaflets in their clinics and more than three-quarters of doctors reported the unavailability of nicotine replacement medications. It would need to be explored further to identify factors leading to the cause of the lack of availability of these materials and medication supports and ways to increase the availability of these in the clinics.

## Strength and limitations

This study was conducted in government healthcare clinics with a large pool of primary care doctors who served as important health care providers to the majority of the Malaysian population. These findings are most relevant and applicable to the current government primary

care clinic settings in Malaysia. However, we only included three districts thus the generalisability of the results needs to be interpreted with caution. The resources in private primary care settings are different from government primary care settings. Thus, the result cannot be applied in private primary care settings. Further studies are needed to explore the challenges faced by private primary care doctors in practising smoking cessation management.

The questionnaire was self-administered and relied on participants' self-reporting answers. As the result, recall and socially desirable bias might happen and there could be a certain degree of over- or under-reporting. The assurance of confidentiality and anonymity of the participants for the survey had been carried out to mitigate such bias. We are unable to determine the good score in advance as lacking report of discriminant validity for the good and poor score. The use of median or mean score might wrongly classify the participants into good attitude or good practice if there were no proficient participants in the study. However, this classification could provide some insight regarding the association between the factors with those in the group of higher and lower scores. The items developed for the assessment of knowledge and attitude components in the questionnaire involved low level of cognitive and affective domains. These limit the validity of the questionnaire in determining the association between knowledge and attitude with the practice of smoking cessation management. The items will need to be revised and strengthened further in future study to improve their validity.

## Conclusion

The practice of smoking cessation assessment and management was poor among primary care doctors. Poor knowledge and attitude, unavailability of nicotine replacement medication and health promotion materials were significant factors associated with poor practice. The intervention targeted at these factors is needed to improve the practice of primary care doctors in delivering quit smoking services in Malaysia.

## Supporting information

**S1 File. Questionnaire for data collection.**
(DOCX)

**S2 File. Factors associated with the practice of smoking cessation management at contemplation phase among primary health care doctors.**
(DOCX)

## Acknowledgments

We would like to thank the Director-General of Health Malaysia for his permission to publish this article.

## Author Contributions

**Formal analysis:** Beatrice Jee Ngee Ling, Ai Theng Cheong.

**Methodology:** Beatrice Jee Ngee Ling, Ai Theng Cheong, Abdul Hadi Abdul Manap.

**Project administration:** Beatrice Jee Ngee Ling.

**Writing – original draft:** Beatrice Jee Ngee Ling, Ai Theng Cheong, Abdul Hadi Abdul Manap.

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
