## [Decision Letter · Decision Letter 0]

26 Jul 2022

PONE-D-22-17659Factors Influencing the Practice of Smoking Cessation Assessment and Management among Primary Care Doctors (SCAAM-DOC)PLOS ONE

Dear Dr.  Beatrice Jee Ngee Ling

Thank you for submitting your manuscript to PLOS ONE. After careful consideration, we feel that it has merit but does not fully meet PLOS ONE’s publication criteria as it currently stands. Therefore, we invite you to submit a revised version of the manuscript that addresses the points raised during the review process.

We look forward to receiving your revised manuscript.

Kind regards,

Billy Morara Tsima, MD MSc

Academic Editor

PLOS ONE

Journal Requirements:

Reviewers' comments:

Reviewer's Responses to Questions

**Comments to the Author**

1. Is the manuscript technically sound, and do the data support the conclusions?

Reviewer #1: Yes

Reviewer #2: Partly

2. Has the statistical analysis been performed appropriately and rigorously? 

Reviewer #1: I Don't Know

Reviewer #2: I Don't Know

3. Have the authors made all data underlying the findings in their manuscript fully available?

Reviewer #1: Yes

Reviewer #2: No

4. Is the manuscript presented in an intelligible fashion and written in standard English?

Reviewer #1: Yes

Reviewer #2: Yes

5. Review Comments to the Author

Reviewer #1: This study explores self-reported practice targeting smoking cessation programmes among primary care physicians in three areas of Malaysia. The areas were selected due to the high density of practices. Of 553 physicians 383 participated, an inclusion rate of appr 69 percent, which is fairly good. The dominating professional group in the material is termed “Medical officer”; the meaning of the term is not explained to the reader. The authors found a lack of knowledge and educational and medical support to assist patients in smoking cessation. They suggest that interventions that target these areas should be provided to doctors in primary care.

The manuscript is well written, with findings that may alter governmental strategies for smoking cessation in Malaysia. The authors clearly state that their findings differ from similar studies in other countries, thus findings and suggestions for improvement is, in my opinion, applicable to Malaysian health care primarily.

I have just a few comments for clarification and improved readability

Title

L1-3m Factors Influencing the Practice of Smoking Cessation Assessment and Management among Primary Care Doctors (SCAAM-DOC)

Should be more specific:

Factors Influencing the Practice of Smoking Cessation Assessment and Management among Primary Care Doctors (SCAAM-DOC) in Malaysia

Abstract:

Results and conclusions are consistent with the text in the main draft.

L40 Repeatedly the authors start a sentence with “Majority” where I would expect a specific form, such as “The majority…” I suggest that the authors consider re-phrasing. The same is observed in L198, 201, 207.

L 43-45 I would suggest that the numbers after comma is restricted to two

L50 Suggested re-phrasing: The knowledge, attitude and practice among primary care doctors in three districts in Malaysia were suboptimal

Introduction

No comments. The aim is fairly good addressed in the results and discussion

Material and methods

Again, restrict numbers to two numbers after comma

L113 The term “medical officer” and “intern” should be defined and described in more detail. In my country a medical officer is a GP mainly working in an administrative role, supporting the national and regional health authorities, and in many cases not in clinical practice at all.

L138 A comma is missing (0800)

L139 Please explain in more detail the evidence for the 5A and 5R components and their role in mapping knowledge and attitude towards smoking cessation in clinical practice. This should include in which clinical setting that the tool has been validated.

Data analysis

I am not a statistician, and not in position to critically review the authors’ choice of analyses. Choosing median and not mean is logical, given that data were not normally distributed. The choice of methods for bivariate and multivariate analyses seems reasonable, but the journal may confer a statistician for a review of this section.

L191 In what way was the ROC relevant for the assessment of pre-contemplation and contemplations. Does this mean that the results from pre-contemplation is more reliable? If so, this should be discussed.

Results

L210-212 “Table 2 shows 64% (246/383) of primary …” and “78% (299/383) reported on the unavailability of nicotine replacement medications” How does Table 2 show these results?

Discussion

I think the authors address their findings, and the relation to existing literature in a reasonable way. They should emphasize that their findings are most relevant to current Malaysian situation because I don’t think that these results necessarily are applicable to other settings.

Strength and limitations

L356 In what way differ “government primary care clinics» from other clinical practices in Malaysia? Does these practices have patients with other sociodemographic characteristics than other practices in the country? If there is a difference, is it relevant to discuss whether different strategies should be applied to the various practices?

L357 Again, consider re-phrasing “majority”: “the major health care providers to a majority of the population”

Conclusions

No comments besides that the authors should emphasize that findings and conclusions are most relevant to the Malaysian setting.

Reviewer #2: 1. Title: it could be better to mention that the study was conducted in 3 districts of Malaysia.

2. It looks like the objectives of the study were 1) to assess knowledge (K), attitude (A) and practice (P) of workers regarding tobacco cessation, and 2) assess the association between P and K,A, other factors.

3. There is a need to include the hypothesis that is (are) tested.

4. Methods:

-What was the study period?

-How were the issues of response bias as participants may have selected response reflecting good practice while in reality they do not practice according to how they responded?

-Were all doctors  involved in the care of patients for smoking cessation?

-How many doctors are in the 3 districts? This should appear in the study setting, or study population section/ paragraph.

-Line 130: "This questionnaire consisted of four domains ..." There are 3 Bloom's taxonomy domains of learning not four. So P of doctors were assessed in smokers at pre-contemplation and contemplation (change model).

- So, the questionnaire consisted of 17 items: 2 for K, 3 for A and 12 for P (10 pre-contemplation and 2 contemplation phase). However, the authors did not clearly state the score allocated to each score. For instance, it is said the K had 2 and A had 6. Does it mean that each item of A had 2 marks while the K items had 1 mark each?

- Use of "norm-referenced score" has limitation. For instance, if there are no proficient participants, some of the participants may be wrongly classify as "good attitude" or "good practice"... It could be better for authors to determine in advance the score that will be classified as "good attitude" or "good practice"...

-There is a need to state the cut-off point of VIF that was used: Did they use the tolerance level of 0.1 (=VIF 10) or 0.2 (=VIF 5).

-Line 188-194: I am not sure why Goodness-of-fit, the Hosmer-Lemeshow test and area under the receiving operating characteristic (ROC) curve were conducted in the study analysis. Authors need to write this clearly.

Results:

-Titles of the tables need improvement: They should be written in the way that even if one removes from the text, they make sense.

-Table 3: The responses are in Likert scale. Putting seldom and never together is confusing.

Discussion:

-One of the major issue I observe is that K is assessed using 2 recall questions. Recall is the lowest level of cognitive. Does it mean that if someone can recall 2 statements by answering "true" that person has a "good knowledge" on smoking cessation?

-The 2 items to assess cognitive domains are low level of bloom's taxonomy (recall). This limits the assessment of this domain and it is hazardous to conclude that psychomotor in pre-contemplation and contemplation is associated with cognitive in smoking cessation management.

-Also, items on attitude enquire the low level of affective which is "Receiving Phenomena" (when using the verb ask). Same as above.

6. PLOS authors have the option to publish the peer review history of their article (what does this mean?). If published, this will include your full peer review and any attached files.

Reviewer #1: **Yes: **Eivind Aakhus

Reviewer #2: No

---

## [Author Response · Author response to Decision Letter 0]

28 Aug 2022

Reviewer #1: This study explores self-reported practice targeting smoking cessation programmes among primary care physicians in three areas of Malaysia. The areas were selected due to the high density of practices. Of 553 physicians 383 participated, an inclusion rate of appr 69 percent, which is fairly good. The dominating professional group in the material is termed “Medical officer”; the meaning of the term is not explained to the reader. The authors found a lack of knowledge and educational and medical support to assist patients in smoking cessation. They suggest that interventions that target these areas should be provided to doctors in primary care.

The manuscript is well written, with findings that may alter governmental strategies for smoking cessation in Malaysia. The authors clearly state that their findings differ from similar studies in other countries, thus findings and suggestions for improvement is, in my opinion, applicable to Malaysian health care primarily.

I have just a few comments for clarification and improved readability

Thank you for the feedback and constructive comments. We had added the details for explaining the term ‘medical officer’ (L117-120) 

Medical officers are doctors who had passed two years of internship training and are involved in the clinical management of patients in primary health care clinics while interns are medical graduates who are still undergoing internship training under the supervision of a specialist.

2. Title

L1-3m Factors Influencing the Practice of Smoking Cessation Assessment and Management among Primary Care Doctors (SCAAM-DOC)

Should be more specific:

Factors Influencing the Practice of Smoking Cessation Assessment and Management among Primary Care Doctors (SCAAM-DOC) in Malaysia

The title has been amended as suggested (L1-3).

Factors Influencing the Practice of Smoking Cessation Assessment and Management among Primary Care Doctors (SCAAM-DOC) in Three Districts of Malaysia

3.Abstract:

Results and conclusions are consistent with the text in the main draft.

L40 Repeatedly the authors start a sentence with “Majority” where I would expect a specific form, such as “The majority…” I suggest that the authors consider re-phrasing. The same is observed in L198, 201, 207.

Amendment done and adjusted. Specific forms “the majority” had been inserted

(L41-42)

The majority of the respondents had poor score of knowledge (62.4%); attitude (58%) and practice (pre-contemplation management:50.9%; contemplation management:75.7%).

(L211-213)

Most of the participants (56%) were aged 31 to 35 years, and the median age was 33 (IQR 4) years. The majority of the respondents (71.5%) was female.

(L220-222)

The majority of the respondents (92.2%) had encountered smokers during consultation in the past one month, and 76.2% of these respondents who had encountered smokers reported that those smokers were willing to quit smoking.

L 43-45 I would suggest that the numbers after comma is restricted to two

Amendment done and adjusted throughout the document

L50 Suggested re-phrasing: The knowledge, attitude and practice among primary care doctors in three districts in Malaysia were suboptimal

Amendment done and adjusted (L51)

The majority of primary care doctors had poor knowledge, attitude and practice of smoking cessation management.

4.Introduction

No comments. The aim is fairly good addressed in the results and discussion

Thank you.

5.Material and methods

Again, restrict numbers to two numbers after comma

Amendments had been done to two numbers after comma throughout the document

L113 The term “medical officer” and “intern” should be defined and described in more detail. In my country a medical officer is a GP mainly working in an administrative role, supporting the national and regional health authorities, and in many cases not in clinical practice at all.

We had added the details for explaining the term ‘medical officer’ (L117-120) 

Medical officers are doctors who had passed two years of internship training and are involved in the clinical management of patients in primary health care clinics while interns are medical graduates who are still undergoing internship training under the supervision of a specialist.

L138 A comma is missing (0800)

Missing comma 0.80 inserted (L146)

L139 Please explain in more detail the evidence for the 5A and 5R components and their role in mapping knowledge and attitude towards smoking cessation in clinical practice. This should include in which clinical setting that the tool has been validated.

We have added the details for clarity. Refer (L146-158)

This questionnaire was validated among 141 primary care doctors from the government health clinics in three districts in the state of Pahang in Malaysia.[16]

The assessment of the knowledge component was on the 5A components (‘Ask’, ‘Assess’, ‘Advice’, ‘Assist’, ‘Arrange’).[16] The two questions mainly assessed the primary care doctors’ familiarity with the sequence and components of the 5As. (Refer S1 files) These were assessed with a dichotomous scale consisting of options of ‘True’ and ‘False’.[16] A score of one was given for a correct answer and 0 score for a wrong answer.[16] 

The attitude component was assessed according to the doctors’ perception on the relevancy of clinical practice guidelines in improving smoking cessation, the worthiness of putting effort in helping smokers to quit smoking and the benefit of repetition in giving advice to assist patients in smoking cessation .[16] 

Data analysis

I am not a statistician, and not in position to critically review the authors’ choice of analyses. Choosing median and not mean is logical, given that data were not normally distributed. The choice of methods for bivariate and multivariate analyses seems reasonable, but the journal may confer a statistician for a review of this section.

L191 In what way was the ROC relevant for the assessment of pre-contemplation and contemplations. Does this mean that the results from pre-contemplation is more reliable? If so, this should be discussed.

Thank you for the feedback

We agreed that the ROC was not relevant for this analysis. We had removed it. 

6.Results

L210-212 “Table 2 shows 64% (246/383) of primary …” and “78% (299/383) reported on the unavailability of nicotine replacement medications” How does Table 2 show these results?

Sorry. There was a typo error and we have deleted the word ‘table2’. (L225-227)

64% (246/383) of primary health care doctors reported the unavailability of smoking cessation banners, brochures and leaflets and 78% (299/383) reported the unavailability of nicotine replacement medications. 

7.Discussion

I think the authors address their findings, and the relation to existing literature in a reasonable way. They should emphasize that their findings are most relevant to current Malaysian situation because I don’t think that these results necessarily are applicable to other settings.

Thank you for the suggestion. We have emphasized this in the section of strength and limitations. 

(L 375-378)

This study was conducted in government primary care clinics with a large pool of primary care doctors who served as important health care providers to the majority of the Malaysian population. These findings are most relevant and applicable to the current government primary care clinic settings in Malaysia.

8.Strength and limitations

L356 In what way differ “government primary care clinics» from other clinical practices in Malaysia? Does these practices have patients with other sociodemographic characteristics than other practices in the country? If there is a difference, is it relevant to discuss whether different strategies should be applied to the various practices?

We had added the differences between government and other clinical practices in Malaysia under Introduction (L75-81 )

The health care system in Malaysia is provided by both the public and private health sectors, and the Ministry of Health being the major provider for the public sector.[7,8] The government healthcare clinics are highly subsidized and accessible to the public.[7,8] The patient only needs to pay RM1 to RM5 (USD 0.30-1.20) for a clinic visit.[7,8] This fee includes consultation, investigations and medications.[7,8] The doctors in government healthcare clinics encounter more chronic disease follow-up cases while in private primary care clinics more acute illnesses were seen.[25] 

We had added the information in the strength and limitation (L379-383)

The resources in private primary care are different from government primary care settings. Thus the result cannot be applied in private primary care settings. Further studies are needed to explore the challenges faced by private primary care doctors in practising smoking cessation management.

L357 Again, consider re-phrasing “majority”: “the major health care providers to a majority of the population”

Amendment done and adjusted (L375-377)

This study was conducted in government primary care clinics with a large pool of primary care doctors who served as important health care providers to the majority of the Malaysian population. 

9.Conclusions

No comments besides that the authors should emphasize that findings and conclusions are most relevant to the Malaysian setting. 

Amendment done and adjusted (L401-405)

The practice of smoking cessation assessment and management was poor among primary care doctors. Poor knowledge and attitude, unavailability of nicotine replacement medication and health promotion materials were significant factors associated with poor practice. The intervention targeted at these factors is needed to improve the practice of primary care doctors in delivering quit smoking services in Malaysia. 

Reviewer #2:

4.Title:

It could be better to mention that the study was conducted in 3 districts of Malaysia.

Thank you for the suggestion. We had amended the title.(L1-3)

Factors Influencing the Practice of Smoking Cessation Assessment and Management among Primary Care Doctors (SCAAM-DOC) in Three Districts of Malaysia

5. Objective

It looks like the objectives of the study were 1) to assess knowledge (K), attitude (A) and practice (P) of workers regarding tobacco cessation, and 2) assess the association between P and K,A, other factors.

The outcome of interest for our study is the practice of smoking cessation management among primary care doctors. Knowledge and attitude are part of the factors that we examined along with other factors such as sociodemographic and the organisation support. 

Thus, we stated our objective as follows:

 ‘The objective of this study was to determine the practice of smoking cessation management among primary care doctors and its associated factors.’ 

(L28-29)

5.Hypothesis

There is a need to include the hypothesis that is (are) tested.

Thank you for the suggestion. We had added the hypothesis. (L94-98)

We hypothesized that the prevalence of practice of smoking cessation management would be low among primary care doctors and there was an association between socio-demographic factors, organizational support, level of knowledge, attitude with the practice of assessment and management in smoking cessation in the pre-contemplation and contemplation phase among primary health care doctors. 

6.Methods:

-What was the study period?

We had added the details in the abstract (L31-34 ) and methods (L106-113) ;

Abstract (L31-34 )

A cross-sectional online survey was carried out among 383 medical officers and interns in all government primary healthcare clinics in the district of Petaling, Klang and Hulu Langat from June to August 2020. All doctors were involved in the care of patients for smoking cessation.

Materials and methods (L106-113 )

These three districts were purposely selected in view of the high density of the population and the large number of doctors served in these areas. There was a total of 32 government primary healthcare clinics comprised of 223 doctors in the district of Hulu Langat, 132 doctors in the district of Klang and 198 in the district of Petaling. The study was conducted from June to August 2020. 

-How were the issues of response bias as participants may have selected response reflecting good practice while in reality they do not practice according to how they responded?

Thank you for the feedback. We acknowledged this issue and the assurance of confidentiality and anonymity of the participants for the survey had been carried out to mitigate such bias (L385-388 )

The questionnaire was self-administered and relied on participants’ self-reporting answers. As the result, recall and socially desirable bias might happen and there could be a certain degree of over- or under-reporting. The assurance of confidentiality and anonymity of the participants for the survey had been carried out to mitigate such bias.

Were all doctors involved in the care of patients for smoking cessation?

Yes. All doctors are involved in the care of patients for smoking cessation.

We had added the information in the section of Study design and Data collection (L110-113)

All the doctors were involved in identifying and counselling smokers to quit smoking during their daily consultation and recruiting those in the contemplation phase to quit smoking clinic. 

-How many doctors are in the 3 districts? This should appear in the study setting, or study population section/ paragraph.

Thank you. Information has been added as suggested. (L108-110)

There was a total of 32 government primary healthcare clinics comprised of 223 doctors in the district of Hulu Langat, 132 doctors in the district of Klang and 198 in the district of Petaling.

-Line 130: "This questionnaire consisted of four domains ..." There are 3 Bloom's taxonomy domains of learning not four. So P of doctors were assessed in smokers at pre-contemplation and contemplation (change model).

Sorry for the confusion. We had changed the word ‘domain’ to ‘component’ based on the terms used by the original authors throughout the document. The four components had been developed by the authors include; knowledge, attitude, practice of smoking cessation management in the pre-contemplation and contemplation phase.

(Aris M, Ehsan S, Mohamed M, Rus R, Jamani N. Reliability and Construct Validity of Knowledge, Attitude and Practice of Medical Doctors on Smoking Cessation Guiedlines. International Medical Journal Malaysia. 2018;17: 199–206. doi:10.31436/imjm.v15i1.1197)

- So, the questionnaire consisted of 17 items: 2 for K, 3 for A and 12 for P (10 pre-contemplation and 2 contemplation phase). However, the authors did not clearly state the score allocated to each score. For instance, it is said the K had 2 and A had 6. Does it mean that each item of A had 2 marks while the K items had 1 mark each? 

The details were added as follows:

Knowledge component (L149-153)

The assessment of the knowledge component was on the 5A components (‘Ask’, ‘Assess’, ‘Advice’, ‘Assist’, ‘Arrange’).[16] The two questions mainly assessed the primary care doctors’ familiarity with the sequence and components of the 5As. (Refer S1 files) These were assessed with a dichotomous scale consisting of options of ‘True’ and ‘False’.[16] A score of one was given for a correct answer and 0 score for a wrong answer.[16] 

Attitude component (L155-164 )

The attitude component was assessed according to the doctors’ perception on the relevancy of clinical practice guidelines in improving smoking cessation, the worthiness of putting effort in helping smokers to quit smoking and the benefit of repetition in giving advice to assist patients in smoking cessation .[16] It was assessed according to the respondents’ degree of agreement with a five point Likert scale of ‘Strongly agree’, ‘Agree’, ‘Don’t know’, ‘Disagree’ and ‘Strongly disagree’.[16] The scores were given as ‘Disagree’ = 1 mark, ‘Strongly disagree’ = 2 marks, and other responses (strongly agree, agree, don’t know) = 0 mark and for the final item about ‘repetition in advising to quit smoking is beneficial’; the reverse coding was applied with the score of ‘Agree’ = 1 mark, ‘Strongly agree’ = 2 marks and other responses (strongly disagree, disagree, don’t know) = 0 mark (Refer S1).[16]

Practice component (L165-171)

The practice component was assessed according to the stage of change: the precontemplation and contemplation phase.[16] The practice of smoking cessation assessment and management for patients at pre-contemplation phase include ‘Ask’, ‘Advice’ and ‘Assess’, followed by the 5R on those who are not ready to quit.[16] The practice of smoking cessation management for those patients at contemplation phase include ‘Assist’ and ‘Arrange’. The items were assessed using a four point Likert scale with the score of ‘Always’ = 2 marks, ‘Frequent= 1 mark, other responses (seldom and never) = 0 mark.[16] 

- Use of "norm-referenced score" has limitation. For instance, if there are no proficient participants, some of the participants may be wrongly classify as "good attitude" or "good practice"... It could be better for authors to determine in advance the score that will be classified as "good attitude" or "good practice"...

We acknowledge the limitations of ‘norm reference score’. However, we were not able to determine in advance the score as there was no report on discriminant validity for this scoring. We have added this in the limitations. (L386-390)

We are unable to determine the good score in advance as lacking report of discriminant validity for the good and poor score. The use of median or mean score might wrongly classify the participants into good attitude or good practice if there were no proficient participants in the study. However, this classification could provide some insight regarding the association between the factors with those in the group of higher and lower scores. 

-There is a need to state the cut-off point of VIF that was used: Did they use the tolerance level of 0.1 (=VIF 10) or 0.2 (=VIF 5).

(L201-203)

There was no multicollinearity detected and the VIF ranged from 1.01 to 6.88. The tolerance level of 0.1 (=VIF 10) was used. 

-Line 188-194: I am not sure why Goodness-of-fit, the Hosmer-Lemeshow test and area under the receiving operating characteristic (ROC) curve were conducted in the study analysis. Authors need to write this clearly.

We had added the information on Hoesmer -Lemeshow test for clarity and removed the ROC curve based on reviewer 1’s comments. (L204-208)

The model fitness was assessed using the Hosmer-Lemeshow goodness of fit test. The analysis with Hosmer-Lemeshow test showed a p-value of more than 0.05, indicating an adequate model fit. 

7.Results

-Titles of the tables need improvement: They should be written in the way that even if one removes from the text, they make sense.

We have edited the titles of Table 2 and 3 to improve clarity

(L237-239)

Table 2. The level of knowledge, attitude and practice of smoking cessation management among primary care doctors

(L258-259)

Table 3 The likeliness of practice among primary care doctors for the items of smoking cessation management in the pre-contemplation and contemplation phase

-Table 3: The responses are in Likert scale. Putting seldom and never together is confusing.

Thank you for the feedback. We had separated those answered ‘seldom’ and ‘never’ to improve clarity. (Page L258-259)

8.Discussion

-One of the major issue I observe is that K is assessed using 2 recall questions. Recall is the lowest level of cognitive. Does it mean that if someone can recall 2 statements by answering "true" that person has a "good knowledge" on smoking cessation?

-The 2 items to assess cognitive domains are low level of bloom's taxonomy (recall). This limits the assessment of this domain and it is hazardous to conclude that psychomotor in pre-contemplation and contemplation is associated with cognitive in smoking cessation management.

-Also, items on attitude enquire the low level of affective which is "Receiving Phenomena" (when using the verb ask). Same as above.

Thank you for the constructive feedback. We acknowledge the limitations of the questionnaires and address this in the limitation. (L393-398)

The items developed for the assessment of knowledge and attitude components in the questionnaire involved low level of cognitive and affective domains. These limit the validity of the questionnaire in determining the association between knowledge and attitude with the practice of smoking cessation management. The items will need to be revised and strengthened further in future study to improve their validity.

---

## [Editor Report · Decision Letter 1]

31 Aug 2022

Factors Influencing the Practice of Smoking Cessation Assessment and Management among Primary Care Doctors in Three Districts of Malaysia (SCAAM-DOC)

PONE-D-22-17659R1

Dear Dr. Ling,

We’re pleased to inform you that your manuscript has been judged scientifically suitable for publication and will be formally accepted for publication once it meets all outstanding technical requirements.

Kind regards,

Billy Morara Tsima, MD MSc

Academic Editor

PLOS ONE

Additional Editor Comments (optional):

The reviewers' comments and suggestions are satisfactorily addressed and the manuscript has been improved.
---

## [Editor Report · Acceptance letter]

21 Sep 2022

PONE-D-22-17659R1 

Factors Influencing the Practice of Smoking Cessation Assessment and Management among Primary Care Doctors (SCAAM-DOC) in Three Districts of Malaysia 

Dear Dr. Ngee Ling:

I'm pleased to inform you that your manuscript has been deemed suitable for publication in PLOS ONE. Congratulations! Your manuscript is now with our production department. 

Kind regards, 

on behalf of

Dr. Billy Morara Tsima 

Academic Editor

PLOS ONE